# Development of Preclinical Ultrasound Imaging Techniques to Identify and Image Sentinel Lymph Nodes in a Cancerous Animal Model

**DOI:** 10.3390/cancers14030561

**Published:** 2022-01-22

**Authors:** Marion Bacou, Vidya Rajasekaran, Adrian Thomson, Sandra Sjöstrand, Katarzyna Kaczmarek, Anna Maria Ochocka-Fox, Adam D. Gerrard, Susan Moug, Tomas Jansson, Helen Mulvana, Carmel M. Moran, Susan M. Farrington

**Affiliations:** 1Cancer Research UK Edinburgh Centre, Institute of Genetics and Cancer, University of Edinburgh, Edinburgh EH4 2XR, UK; m.bacou@sms.ed.ac.uk (M.B.); vrajasek@exseed.ed.ac.uk (V.R.); 2Centre for Cardiovascular Science, University of Edinburgh, Edinburgh EH16 4JT, UK; Adrian.Thomson@ed.ac.uk; 3Department of Biomedical Engineering, Faculty of Engineering, Lund University, 223 63 Lund, Sweden; sandra.sjostrand@bme.lth.se; 4Biomedical Engineering, University of Strathclyde, Glasgow G1 1QE, UK; katarzyna.kaczmarek@strath.ac.uk (K.K.); Helen.Mulvana@strath.ac.uk (H.M.); 5MRC Human Genetics Unit, Institute of Genetics and Cancer, University of Edinburgh, Edinburgh EH4 2XR, UK; aochocka@exseed.ed.ac.uk (A.M.O.-F.); Adam.gerrard@ed.ac.uk (A.D.G.); 6Department of Surgery, Royal Alexandra Hospital, Paisley PA2 9PJ, UK; Susan.Moug@ggc.scot.nhs.uk; 7Department of Clinical Sciences Lund, Biomedical Engineering, Lund University, 223 63 Lund, Sweden; tomas.jansson@med.lu.se; 8Clinical Engineering Skåne, Digitalisering IT/MT, Skåne Regional Council, 291 89 Lund, Sweden

**Keywords:** contrast-enhanced ultrasound, 3D ultrasound, preclinical, lymph node, colorectal cancer, metastatic mouse model

## Abstract

**Simple Summary:**

Bowel cancer is the fourth most common cancer in the UK. Treatment is dominated by major surgery because current imaging modalities cannot accurately determine lymph node involvement or vascular invasion. Although potentially curative, surgery carries a high risk of short- and long-term morbidity, including stoma formation. Optimized pre-treatment imaging would decrease the number of bowel cancer patients requiring major surgery. Such imaging would also be equally applicable to other cancers where local resection could significantly improve patient quality of life without compromising long-term outcomes (e.g., melanoma, head and neck cancers, gastro-esophageal, bladder). In this study, we created two mouse models (tumor and control) and used the resolution of high-frequency ultrasound imaging and parameters calculated from dynamically contrast-enhanced ultrasound to predict the likelihood of draining lymph nodes to be involved in the disease.

**Abstract:**

Lymph nodes (LNs) are believed to be the first organs targeted by colorectal cancer cells detached from a primary solid tumor because of their role in draining interstitial fluids. Better detection and assessment of these organs have the potential to help clinicians in stratification and designing optimal design of oncological treatments for each patient. Whilst highly valuable for the detection of primary tumors, CT and MRI remain limited for the characterization of LNs. B-mode ultrasound (US) and contrast-enhanced ultrasound (CEUS) can improve the detection of LNs and could provide critical complementary information to MRI and CT scans; however, the European Federation of Societies for Ultrasound in Medicine and Biology (EFSUMB) guidelines advise that further evidence is required before US or CEUS can be recommended for clinical use. Moreover, knowledge of the lymphatic system and LNs is relatively limited, especially in preclinical models. In this pilot study, we have created a mouse model of metastatic cancer and utilized 3D high-frequency ultrasound to assess the volume, shape, and absence of hilum, along with CEUS to assess the flow dynamics of tumor-free and tumor-bearing LNs in vivo. The aforementioned parameters were used to create a scoring system to predict the likelihood of a disease-involved LN before establishing post-mortem diagnosis with histopathology. Preliminary results suggest that a sum score of parameters may provide a more accurate diagnosis than the LN size, the single parameter currently used to predict the involvement of an LN in disease.

## 1. Introduction

Colorectal cancer (CRC) is the second most common cause of cancer death worldwide, representing 940,000 deaths in 2020 [1]. CRC, as other carcinomas, is believed to spread to distant organs through the lymphatic system; hence, lymph node (LN) status is a crucial prognostic factor that determines optimal treatment planning [2,3]. Despite producing high-quality images of primary tumors, magnetic resonance (MR) and computed tomography (CT) scanners, routinely used in the UK, lack precision for certain assessment and diagnosis of LNs, often leading surgeons to perform major resection, which involves removing large parts of the intestines and healthy surrounding LNs, to avoid recurrence of the disease [4,5]. A more confident assessment of an LN’s status could lead to a more personalized approach to surgery and further oncological treatment. Limiting surgical resection would significantly reduce surgical complications and consequently result in an improved quality of life for the patient. Grey scale ultrasound imaging (US) has previously been used to identify and characterize the LNs adjacent to the primary tumor [4,6,7,8]. Specifically with respect to rectal cancer, a meta-analysis of clinical studies undertaken between 1985 and 2005 demonstrated that endoluminal ultrasound was at least as good as MRI or CT for identification of nodal disease and that no significant difference between imaging modalities was measured with respect to the staging of nodal status [4]. Lin et al. demonstrated that quantitative parameters such as short axis length, echogenicity, shape, and how distinct the lymph node margin were independent features capable of predicting lymph node metastasis in lung cancer patients [7]. In a highly detailed paper, Walk et al. demonstrated the efficacy of high-resolution ultrasound for imaging and accurately reconstructing small structures such as mouse cervical lymph nodes in 3D [6].

US has the advantage of being more cost-efficient than CT and MRI and, in addition, does not involve the use of ionizing radiation. Combined with color Doppler and ultrasonic contrast agents, it can help determine the vascularization and the shape of superficial LNs [9]. Unlike CT and MRI, US provides real-time images and can therefore be utilized for fine-needle aspiration biopsies [6,10]. Despite promising findings, EFSUMB guidelines do not recommend the use of US or CEUS for the characterization of LN for clinical use; however, they encourage research in this field [11]. High-resolution US imaging, using ultrasound transducers with center frequencies up to 70 MHz, enables the anatomy and physiology of small animal models to be assessed in real-time and with spatial resolution better than 50 μm [12]. This imaging technique has become an increasingly versatile tool within biological research facilities and is widely used in longitudinal studies where data acquired at multiple timepoints can provide data on the size of organs and the magnitude of blood velocity, which can be used to inform our understanding of the progression of diseases.

Ultrasonic contrast agents are composed of encapsulated microbubbles and aid visualization of vascular structures and quantification of blood flow. Commercially available contrast microbubbles for clinical applications are lipid-encapsulated bubbles and have either fluorocarbon or sulfur hexafluoride (SF_6_) gas encapsulated within them. Optimal imaging of these microbubbles relies upon detection and preferential display of the non-linear signal generated from these microbubbles when insonated with ultrasound waves of sufficient magnitude and at the resonant frequency of the bubbles. At the higher frequencies utilized for preclinical studies, the majority of the bubbles designed for the clinical application do not resonate. MicroMarker microbubbles (Fujifilm) have been specifically designed for such high-frequency applications, and the non-linear signal scattered from these microbubbles can be detected, displayed, and used to study and quantify vascular dynamics in small animals.

In this study, we used intestinal tumor cells with proven ability to induce metastasis [13,14] and developed a simplified and refined method of inducing an LN metastasis mouse model of CRC that exhibits spread to LNs, and we examined it using 3D-US and CEUS in order to create a scoring system capable of predicting LN involvement in the disease.

## 2. Materials and Methods

All animal procedures were performed at a University of Edinburgh’s Biomedical and Veterinary Sciences facility under UK Home Office project license P02F16F82. This research was undertaken in accordance with the Animals Scientific Procedures Act (ASPA) 1986 Amendment Regulations 2012, following ethical review by the University of Edinburgh Animal Welfare and Ethical Review Board (AWERB). From 8- to 12-week-old C57BL/6J mice were purchased from Charles River UK and maintained in groups of 2–5 mice in individually ventilated cages.

In this study, we aimed to induce LN metastases in healthy recipients by injecting intestinal tumor organoids issued from *villin*Cre^ER^ *Kras*^G12D/+^ *Trp53*^fl/fl^ *Rosa26^N1icd^*^/+^ (KPN) animals in the left hind leg (KPN-injected mice labeled 1, 2, 3, 4, and 5) [14]. Right-sited inguinal LNs of each mouse were used as internal controls. Organoids produced from normal intestinal tissue of wild-type C57BL6J animals (WT) were also injected into the left hind leg of three other mice (WT-injected mice referred to as A, B, and C), and these left-sited inguinal LNs were used as a negative control. Right-sited inguinal LNs of each mouse were used as internal controls. We then utilized 3D ultrasound and CEUS to assess the volume, morphology, and flow dynamics of left and right inguinal LNs in vivo. These parameters were then used to create a scoring system to predict the likeliness of an LN to be diseased. Further details are given below.

### 2.1. Tumor Induction and Organoid Production

Tumor organoids were generated by the Beatson Institute, Glasgow, from intestinal tumors of tamoxifen-induced KPN mice and had previously been extensively characterized for induction of intestinal cell metastases [14]. Control organoids from normal intestines of C57Bl6/J mice were generated in-house. Organoids were cultured in vitro with mADF+++ (DMEM/F-12 medium (Dulbecco, ThermoFisher Scientific) supplemented with 2 mM Glutamax (Fisher Scientific), 10 mM HEPES (N-2-hydroxyethylpiperazine-N- 2-ethane sulfonic acid, Fisher Scientific), 1.5 mM BSA (bovine serum albumin, Sigma-Aldrich), 100 U/mL penicillin/streptomycin, and 1 mM NAC (N-acetyl-L-cysteine, Sigma-Aldrich)) supplemented with 10% Noggin in-house conditioned medium (using HEK293 cells stable transfected with pcDNA3 NEO mouse C-terminal Noggin), 20% R-spondin in-house conditioned medium (using HEK293-HA-RspoI-Fc cells), 50 μg/mL mouse EGF recombinant (epithelial growth factor, Thermo-Scientific), 1× B27 supplement (Invitrogen), and 1× N2 supplement (Invitrogen).

### 2.2. Cell Injection and Animal Monitoring

The number of animals required for this study was estimated using the NC3Rs’ power calculation tool (issued from R 3.5.2 and the package power.*t*.test; R Core Team (2021). R: A language and environment for statistical computing. R Foundation for Statistical Computing, Vienna, Austria). Initial power calculations indicated that a minimum of 3 animals were needed per group. A total of 5 animals (3M, 2F) and 3 animals (1M, 2F) were injected in the left hock with approximately 150 disrupted KPN or WT intestinal organoids, respectively, using the technique described by Kamala in 2007 [15] (Figure 1). The injection site was monitored during and after injection, and any occasional reflux of organoids suspension was noted. The right hock of each animal was not injected and was used as an internal control. Mice were visually checked daily for signs of pain, distress, and/or tumor bearing and weighed weekly. Endpoint (via euthanasia) was determined when the leg tumor measured using calipers was over 1.2 cm in diameter (as per The University of Edinburgh Standard Operating Procedures and UK Home Office guidelines) or if signs of pain, distress, difficulty in breathing, or skin breakage were observed. US scans were performed up to 4 times per animal.

### 2.3. Ultrasound Experiments and Analysis

US data were acquired using the Vevo3100 preclinical ultrasound scanner (FUJIFILM VisualSonics, Toronto, ON, Canada).

#### 2.3.1. Animal Preparation

Ultrasound scanning was carried out under isoflurane anesthesia using 100% O_2_ at a rate of 1 L/min with an initial induction set at 4% isoflurane, then maintained at 2%. Mice were laid supine on a controlled heated table, and respiration, heart rate, and temperature were monitored at all times with electrodes and a rectal thermometer. Leg and ventral hair were removed with clippers and epilating cream, and pre-warmed US transmission gel (Aquasonic 100, Parker Laboratories, Fairfield, NJ, USA.) was applied to the lower abdominal quadrants to scan both left and control inguinal LNs. Special care was taken to avoid excessive pressure when scanning to avoid compression of the LN. Both left (organoids-draining) and right (control) inguinal LNs of each mouse were scanned.

#### 2.3.2. 3D-US

3D-US data were generated by imaging LNs using the 40 MHz center frequency MX550D US probe with the image optimized so that at least one focal position lay at the same depth of the LN location and the LN was centered in the scan-plane of the transducer. Transmission power was 100%. Respiratory gating was turned on to reduce motion generated by organ movements deleterious for 3D data acquisition. The acquisition depth was adjusted for each animal, depending on the size of the overlaying fat capsule. The transducer was mounted on a 3D motor set up at a minimal step size of 0.038 mm, allowing the acquisition of consecutive B-mode images that were later used offline to reconstitute 3D volumes with Vevo^®^ LAB software. To do so, the region of interest (ROI) was manually drawn at the borders of the LNs to be analyzed, and the software then automatically assimilated the frames into a 3D reconstitution and performed volumetric quantification of the LN.

The 3D reconstitutions of the LNs were used to assess and give a score of either zero or one to the following parameters:Volume ratios: For each animal, the volumes of the left inguinal LNs were compared to the inguinal LNs from the control right leg. Left LNs that were significantly larger (≥5×; *p* < 0.05) than the control right LNs were given a score of 1 (Table 1).Shape: The length was determined as the longest longitudinal diameter found on the 3D US data, and the width was the longest perpendicular diameter along with the same 2D B-mode image as the long axis (Figure 2). An LN was defined as round when the length to width ratio was under 2. Round LNs were given a score of 1, whilst non-round LNs were given a score of 0 (Table 1).Hilum: The 3D reconstitutions of the LNs were assessed for the presence of echogenic hilar structures. The hilum forms a depression on the surface of LN where the branching of the efferent lymphatic and blood vessels pass (Figure 1). LNs where no hilum was detected were given a score of 1, and LNs where a pedicle was detected were given a score of 0 (Table 1).

#### 2.3.3. CEUS

Non-Targeted MicroMarker™ (MM) (FUJIFILM, VisualSonics) contrast agent was resuspended in saline following the manufacturer’s instructions, and 50 μL of contrast agent was introduced as a bolus into the tail vein via a 27 g butterfly catheter. All injections were performed by the same user, and suspensions were delivered at a steady rate. For each animal, the left inguinal LN was scanned first. Once the dynamic contrast enhancement data had been collected, several high acoustic power output burst sequences were used until enhancement within the LN had returned to baseline values. A second contrast injection was then performed to measure the dynamic contrast enhancement within the right control inguinal LN.

For CEUS imaging, the MX250 US transducer with a nominal center frequency of 21 MHz was used. Although the spatial resolution was reduced with this probe compared to the higher frequency MX550D probe, all LNs could be visualized. For each LN, the probe was positioned such that the largest longitudinal extent of the LN was centered in the B-mode image. Contrast-enhanced imaging mode on the Vevo3100 was activated prior to injection of the contrast. In this mode, non-linear imaging of the contrast agent is utilized using amplitude modulation techniques. The settings and magnitude of the parameters that affect the magnitude of the contrast data (TGC settings, contrast gain, focal position, output power, line density, persistence, and beam-width) were initialized and recorded to ensure similar set-ups for future scans. Cine loops of 1000 frames that encompassed acquisition before, during, and after bolus injections into an anesthetized animal were acquired.

Bolus perfusion CEUS analysis was performed offline using Vevo^®^ CQ software. For data analysis, the ROI (i.e., the margins of the LNs) was drawn free-hand on one representative frame. The automatic Motion Correction tool was applied using this image as a reference to realign every frame that shifted due to motion related to the organ (e.g., respiration) or slight probe movements. The cine loop was then manually checked, and any frame misaligned was excluded from further analysis. A data-fitting curve was then auto-generated by the software, and the following parameters were analyzed: Time-to-peak (TTP): TTP is a perfusion parameter that measures the period from the first appearance of the contrast in the ROI to the time of highest enhancement. A significantly lengthened TTP in the left LN compared to the right control LN was given a score of 1, whilst a reduced TTP in the left LN was given a score of 0 (Table 1).Wash-in rate (WiR): WiR is determined as the slope of the linear fit to the enhancement data from the time between contrast agent arrival in the ROI and the peak enhancement. It is used to determine the kinetics of the contrast agent inside of the ROI analyzed. Significantly smaller WiR in the left LN compared to the right LN was given a score of 1, whilst a higher WiR was given a score of 0 (Table 1).

### 2.4. Histopathology

Dissection of the mouse was performed immediately after termination. LNs were collected into 10% neutral buffered formalin solution (NBF, Sigma-Aldrich) and paraffin-embedded for further examination and assessment of the size, shape, and presence of intestinal metastases.

Then, 5 μm thick slices were stained with hematoxylin and eosin (H&E) or with antibodies, scanned with Nanozoomer Whole Slide Imager (Hammasatu), and analyzed using NDP Servu software.

LNs were stained with epithelial marker Cytokeratin 7 (Abcam ab181598, 1/500). The slides were deparaffined by incubating them in a series of xylene and ethanol baths. The endogenous peroxidase activity was blocked using 3% H_2_O_2_ in PBS, and the heat-induced epitope retrieval (HIER) method was performed with sodium citrate 0.1% Tween 20× for 10 min. Permeabilization was performed with 0.5% Triton X in PBS for 20 mins, and non-specific antigenic sites were blocked by using 5% donkey serum for 1 h. Cytokeratin 7 was added overnight at 4 °C, and a secondary antibody was added for 2 h at room temperature (RT). DAB was added for 10 mins at RT, and slides were counterstained before being dehydrated and mounted.

### 2.5. Statistical Analysis

All five parameters of the inguinal left LNs were compared to the control inguinal LNs from the same mouse using paired *t*-tests on Prism (version 8.0.0 for Windows, GraphPad Software, San Diego, CA, USA, www.graphpad.com). Significance was considered for a *p*-value of less than 0.05.

## 3. Results

### 3.1. All Mice Injected with KPN Organoids Develop Epithelial Solid Tumors

Animals injected with KPN intestinal organoids develop hind leg tumors visually detectable from 4 weeks post injection (5.9 weeks average). Data indicated that leg tumors could be detected as early as 2.5 weeks post injection with the 3D-US technique (3.4 weeks average) (Figure 3). We were able to confidently distinguish a tumor from muscular tissue when the leg tumor was as small as 54 mm^3^. Leg tumors reaching 1.2 cm in size (diameter measured externally with a caliper) were the endpoint factor for all KPN-injected animals, which otherwise did not present any sign of distress or pain.

No animal injected with WT intestinal organoids presented any sign of distress, pain, or tumor bearing at any timepoint.

### 3.2. 3D-US Data Show Morphology Alterations in the Left LNs of KPN-Injected Mice

#### 3.2.1. Volume

Size is currently the most accepted method to determine LN status, as this parameter can be assessed using US, MRI, and CT [16,17]. A larger-than-average LN is generally considered to be harboring metastases. Clinical studies usually include a size threshold for which an LN is considered pathological [7,17]. With little information about normal LN size in mice available in the literature, it was decided that for each animal, the volume of the tumor-draining inguinal LN would be compared to the uninvolved (control) inguinal LN draining the right leg.

Left inguinal LNs of WT-injected mice were not significantly larger (*p* = 0.726) than the control (right) LNs, with an average size of 1.384 ± 0.264 mm^3^ and 1.379 ± 0.333 mm^3^, respectively (Figure 4a).

However, 3D-US scans performed at different timepoints showed that the volumes of the left inguinal LNs of KPN-injected mice were increasing as the tumors were developing. The left LNs of all KPN-injected animals were statistically larger (*p* = 0.0143) than the control (right) ones from 4 weeks post injection. At the endpoint, the left inguinal LNs volumes of the KPN-injected mice were on average 9.92 times larger than the inguinal LNs of the other hind leg (control). The average volume of the left and right control LNs of the KPN-injected mice were 12.076 ± 1.281 mm^3^ and 1.438 ± 0.691 mm^3^, respectively (Figure 4b,c). The significantly larger left inguinal LNs of KPN-injected suggests the mice were housing metastases, whilst the left inguinal LNs of the WT-injected mice were tumor-free. Left LNs of WT organoid-injected mice were not significantly larger than the control (right) LNs, suggesting that the injection of non-tumor organoids does not have any influence on the size of the draining LNs.

#### 3.2.2. Shape

The shape is a parameter sometimes utilized for the characterization of LNs in cancer, where deviation towards a more rounded morphology is associated with malignant involvement. LNs are considered to be round and, therefore, more likely to be malignant if the length to width ratio of a representative 2D image is less than 2 [17,18]. 

Left and control (right) inguinal LNs of the WT-injected animals did not have a significantly different shape with respect to rounded morphology (*p* = 0.179). Only one WT-injected animal (Mouse A) had its left inguinal LN considered as round, with a length to width ratio of 1.901.

At the endpoint, US images demonstrated that the left LNs of the KPN-injected mice had significantly different and rounded shapes compared with the control (right) LNs (*p* = 0.0002). All left LNs of KPN-injected mice were morphologically round, with an average length to width ratio of 1.621 ± 0.181, whilst all the control LNs were considered as non-round with an average ratio of 2.432 ± 0.166 (Figure 5).

#### 3.2.3. Hilum

The LN hilum, also called the pedicle, is an indent where the efferent lymphatic vessels and blood vessels pass (Figure 4c). A loss of detectable hilum can be indicative of metastatic spread [7,17,19]; thus, an ability to distinguish the characteristic indent of a healthy hilum using US was examined in each mouse. The hila of the left LNs of WT-injected animals were all visible. Conversely, none of the hila in the left inguinal LNs of the five KPN-injected animals were detected, suggesting metastatic spread in these structures.

### 3.3. CEUS Data Show Left LNs of KPN-Injected Mice Have a Reduced Blood Flow

CEUS imaging was performed immediately prior to euthanasia on all animals of the study such that quantitative analysis of the contrast enhancement could be undertaken. Two parameters of interest were assessed: wash-in rate (WiR), which aims to quantify organ perfusion [19,20] and where a low WiR value indicates poor perfusion [21], and time-to-peak (TTP), which provides an assessment of blood flow [20]. 

CEUS data for all WT-injected mice (A, B, and C) were successfully analyzed; however, data from one KPN-injected animal (Mouse 5) could not be investigated due to Vevo^®^ CQ software being unable to perform satisfactory motion correction on the control inguinal LN, resulting in the loss of too many data frames.

For each mouse and for each LN (left and right), WiR was calculated over an ROI which encompassed the largest cross-sectional area of the LN. To assess the difference in WiR between left and right LNs, a ratio was calculated such that the rate was normalized to that of the control (right) LN. Using this ratio, the WiR did not indicate any significant difference between the left and right LNs of WT-injected mice.

In KPN-injected animals, the WiR of left LNs was not significantly different (*p* = 0.768) when all four KPN-injected animals analyzed were included (Figure 6a). However, when Mouse 4 was excluded from analysis, as the contrast-enhancement pattern in its LN was radically different to the ones in the other mice, the WiR of the left LNs in the remaining three KPN-injected animals were significantly smaller (*p* = 0.015) than the control (right) LNs (Figure 6b). This suggests that, in general, the left LNs are less perfused than the control inguinal LNs. Conversely, the left LN of Mouse 4 is more perfused than its control LN.

Time-to-peak (TTP) is a parameter correlated to blood flow. Its value is extracted from the fitted curve generated by the Vevo^®^ CQ software from the contrast enhancement data [21]. However, literature remains inconclusive about the interpretation of TTP. Several studies indicate that shorter TTP is usually seen in tumor-bearing LNs due to the high vascularization of spontaneous tumors [22,23]. However, a reduced blood flow can occur in the LNs when internal or surrounding vessels are overly compressed, for example, by a US transducer or by a tumor, and as such, this would prolong the TTP [19].

Data indicated that the TTP measurements for CEUS imaging of the left LNs of KPN-injected mice were significantly longer than of control (right) LNs (*p* = 0.0282), suggesting that the blood flow is reduced in the left LNs of this mouse model (Figure 7). Left LNs from KPN-injected mice had an average TTP of 4.01 ± 1.26 s, and control (right) LNs had a TTP average of 1.93 ± 0.77 s. Special care was taken to avoid compression of the scanned regions with the transducer during data acquisition, suggesting that the blood vessels within or leading to the left LNs of the KPN-injected mice are being compressed by the anatomy of the mouse rather than by external forces.

### 3.4. Development of a Scoring System to Predict the Left LNs Status

The score associated with each mouse LN, assessing metric volume, shape, evidence of hilum, TTP, and WiR, was integrated into a scoring system as described in Section 2.3.2 and Section 2.3.3 (Table 2). A larger sum score suggested a greater probability for the tumor-draining LN to be malignant.

### 3.5. Post-Mortem Histopathology Confirms the Presence of Metastases in Left LNs of KPN-Injected Mice and Was Used to Generate Predictive Values, Sensitivity, and Specificity

Histopathology was performed on every inguinal LN to allow assessment of size, shape, and presence of tumor cells. Antibody targeting Cytokeratin 7 was utilized to detect epithelial tumors produced by the injected intestinal organoids.

In WT-injected mice, no obvious difference was observed between the left and control (right) inguinal LNs. Mouse A’s left LN was assessed as non-round at pathology report, despite having been characterized as round with 3D-US analysis. No metastases were detected in left LNs of WT-injected mice.

Data indicated an increase in volume and a modification of the shape of all left inguinal LNs of KPN-injected mice, confirming observations made in vivo with US. Immunostaining using antibody targeting Cytokeratin 7 confirmed that 80% (4/5) of the tumor-draining inguinal LNs present with detectable epithelial metastases (Figure 8). No metastasis was detected in the tumor-draining LN of Mouse 4, potentially explaining the difference of WiR in this sample compared to the tumor-draining LNs of the other KPN-injected mice.

Histopathology results were used to calculate the diagnostic performance and accuracy of each of the metrics examined and sensitivity, specificity, positive predictive value (PPV), and negative predictive value (NPV) for these individual and combined parameters when the sum score was total (100%) or partial (80%, 60%, 40% or 20%) (Table 3). The diagnostic performance data indicate that the CEUS measured parameters TTP and WiR are comparable to the 3D parameters volume/shape and hilum. WiR appears to actually improve the diagnostic accuracy and PPV compared to the 3D parameters and TTP.

When the sum score positivity threshold was maximal (100%), and all parameters coexisted, then there was improved diagnostic accuracy (100%), specificity (100%), and PPV (100%-synonymous with precision), whereas when 40–80% of parameters coexisted, they give comparable sensitivity, specificity, PPV, and NPV of 100%, 75%, 80%, and 100%, respectively, with a diagnostic accuracy of 88%.

## 4. Discussion

In this study, a mouse model of CRC with LN metastasis was successfully created by injecting KPN intestinal tumor organoids into the hock of C57BL6J mice. All KPN-injected mice developed solid leg tumors that were visually detectable from 4 weeks post injection. Indeed, the size of the primary tumor made subsequent blinded analysis impractical. The control mice injected with WT intestinal organoids did not develop tumors.

The hock and foot area is known to be drained by both popliteal and inguinal LNs [24]. In this pilot study, we focused our experiments exclusively on the inguinal LNs due to the complications that would be inevitable were we to examine the popliteal LN due to its proximity to the primary leg tumor. Indeed, at the late stages of this experiment, the primary leg tumor was so large (up to 810 mm^3^) that it often engulfed the popliteal LN, making its detection with US difficult and often impossible.

The inguinal LNs of each mouse were observed with US and images interpreted with the aid of five analytical parameters allowing objective assessment of LN presentation and the development of a scoring system that was organized into a table summarizing each LN. Whilst size is a widely accepted single parameter used to determine LN status when MRI and CT are employed, it is also acknowledged that when used alone, size has limited accuracy [25]. To increase the sensitivity of US imaging as a diagnostic tool, the shape, absence of hilum, and blood flow of the tumor-draining LNs were also assessed, in addition to their volumes.

LN volumes were calculated from a series of ultrasound B-mode images acquired through an automatic translation of the ultrasound probe mounted on a translation stage to traverse along the long axis of the LN at predetermined intervals. To enhance sensitivity to volume alterations in LNs, which in our study ranged in value from 0.75 to 13.18 mm^3^, ultrasound scans were acquired using the smallest step size available to calculate the 3D-US data. Although this had implications on scanning time (up to 3 min per structure), it ensured that the most precise 3D volumes were obtained.

According to our observations on control (right) inguinal LNs, size varied across individuals of the same age. In a 2017 study, Tuner and Mabbott reported that murine LNs undergo structural modifications as the individual ages [26]. To minimize the impact of these age-related differences in our data, we only performed statistical analyses on data collected on the same day from the left and control (right) inguinal LNs of individual mice. Within this study, experimental animals were euthanized at different ages, depending on when the endpoint was reached (14–24 weeks of age, average 17.5 weeks). To further counter these biological variations and allow data analysis, volume data were normalized across animals.

All morphology parameters (volume, size, and presence of hilum) performed with US suggested the presence of metastasis in the tumor-draining LNs of each of the KPN-injected mice analyzed. Conversely, histopathology examinations confirmed the presence of metastases in only 80% (4/5) of the tumor-draining LNs. Although admittedly, a micro-metastasis may have been missed during the pathology examination, these data rather indicate that the sole use of B-mode ultrasonic assessment of the morphology of the LN to determine its involvement in cancer is not infallible and could lead to an erroneous false-positive diagnosis. In addition, control Mouse A’s left LN was falsely assessed as round and therefore potentially metastatic with US but was diagnosed as healthy upon pathological examination.

CEUS was also performed at the endpoint to assess the blood flow inside the LNs of interest with time-to-peak (TTP) and wash-in rate (WiR) used in this study. Whilst studies examining LNs usually perform intra- or peri-tumoral injections of US contrast agents [9,22,27], injection via the tail vein led to the most enhancement in our study. Jafarnejad et al. reported that murine LNs contained several thousand blood vessels with diameters ranging from 4 to 82 μm, indicating that US contrast agents, such as MicroMarker^®^ of mean diameter 1.8 μm, can indeed reach and enhance such organs when injected via the bloodstream [28,29].

Literature indicates that shorter TTP is usually seen in tumor-bearing LNs due to the neovascularization produced by most spontaneous tumors [22,23]. However, the longer TTP observed in all the tumor-bearing LNs in our study may be due to the lack of neovascularization generated by the allograft tumor mouse model utilized. CEUS experiments indicated an absence of enhancement and poor vascularization (<3%, *n* = 5 mice) in the primary leg lesion, justifying the long TTP in the LNs harboring tumor metastases.

With the exception of Mouse 4, WiR data were consistent with the TTP results, reinforcing the theory that the blood flow was reduced in the tumor-draining LNs as compared to the control (right) ones. Histopathology indicated that, unlike that of other KPN-injected animals, the left LN of Mouse 4 was not harboring any obvious metastasis, possibly explaining the different WiR observed for this animal.

Examination of the diagnostic performance of each of the US-studied parameters was assessed against pathology. Amongst each of the individual parameters, all demonstrated >65% prediction of LN metastatic involvement (PPV = 67–100%); however, due to the large confidence intervals, a weighed approach of each parameter was excluded. 

A sum score (score positivity threshold) of equally weighted parameters was then created, where higher scores indicated a higher likelihood for the analyzed LN to be harboring metastases. For a 100% total sum score of the combined parameters, the diagnostic accuracy, PPV, and specificity was 100%, and the combined assessment to predict the likelihood of metastasis was improved in comparison to other scores individually—for example, compared to sensitivity, specificity, PPV, NPV, and diagnostic accuracy of the volume parameter alone, whose values are 100%, 75%, 80%, 100%, and 88%, respectively.

It should be noted that within this pilot study, although there was adequate power, relatively small groups of animals were involved. The data presented would certainly be strengthened with greater sample sizes, with likely improvement in the confidence intervals leading to more reliable statistics. Such future work would potentially improve the scoring system and provide insight into which individual parameters may become a focus to continue to improve diagnostic accuracy going forward. In the current dataset, WiR provided the strongest diagnostic indication, with improved diagnostic accuracy and PPV. Access to a larger data set would support the development of a weighted scoring system that would enhance diagnostic accuracy. However, it remains the case that within this study, the values obtained for sensitivity, specificity, PPV, and NPV provide a clear indication of how each individual parameter, and indeed the sum of their scores, performs in terms of efficacy in predicting metastasis in the LNs [30].

The use of scoring tables to extract or improve the diagnostic value of aggregated prediction tools holds precedent across medicine; for example, the likelihood of prostate cancer, or cancer progression, is commonly indicated through a nomogram combining clinical T stage, serum prostate-specific antigen (PSA) and Gleason scores [31]. Meanwhile, and more illustrative for the purposes of this work, quantification of liver perfusion characteristics via CEUS is commonly applied within the clinical pathway to assess liver disease and has driven the development of the sort of analysis software now routinely available on clinical ultrasound scanners and utilized in this work [32,33].

Further research to complement the data obtained during this pilot experiment would generate more precise sensitivities, specificities, and predictive values, enabling the further refinement of the scoring system; however, these preliminary results already indicate that sum scores of parameters can provide a more accurate diagnosis than a single parameter can be expected to deliver alone.

## 5. Conclusions

This pilot study demonstrated that parameters calculated from B-mode US and CEUS data using high-resolution ultrasound are valuable instruments for predicting murine LNs’ involvement in cancer. Preliminary results demonstrated that LNs with higher sum scores, that is to say, LNs with larger volume, round shape, absence of detectable hilum, and a comparatively reduced blood flow have a higher probability of harboring metastases. Compared with current imaging techniques such as CT and MRI only assessing the size of the LN, the diagnostic accuracy of this pilot study was improved. Further research with more tumor-bearing animals is required to validate and improve the findings and confidently determine if one parameter provides a better prediction of LN metastasis. However, it is believed that the use of a sum score rather than individual parameters would better predict metastatic LNs and could therefore help improve surgical and oncological treatment planning.

## Figures and Tables

**Figure 1 cancers-14-00561-f001:**
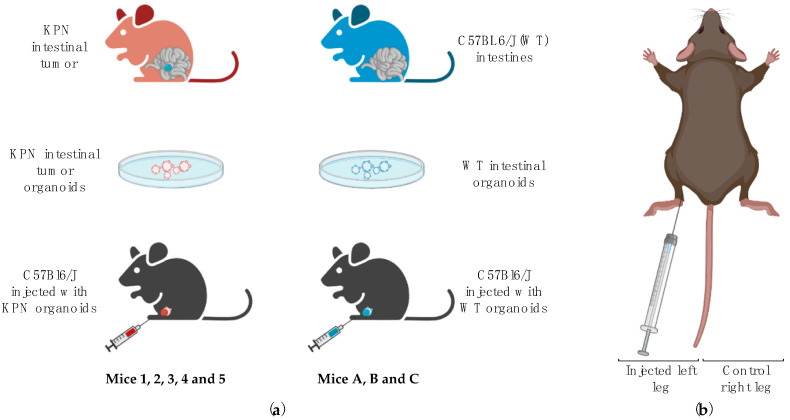
(**a**) KPN [13,14] or WT organoids were generated from KPN or C57BL6/J mice, respectively, and were injected into the hind leg of C57BL6/J mice. Animals injected with KPN intestinal tumor organoids are referred to as KPN-injected animals (Mice 1, 2, 3, 4, and 5), and mice injected with WT intestinal organoids are labeled WT-injected animals (Mice A, B, and C); (**b**) Each mouse was injected with ~150 disrupted KPN or WT organoids resuspended in 50 μL complete medium. Injections were performed in the hock of the left hind leg. The right leg of each mouse was not injected and was used as an internal control. Images generated using BioRender.com (accessed on 4 January 2021).

**Figure 2 cancers-14-00561-f002:**
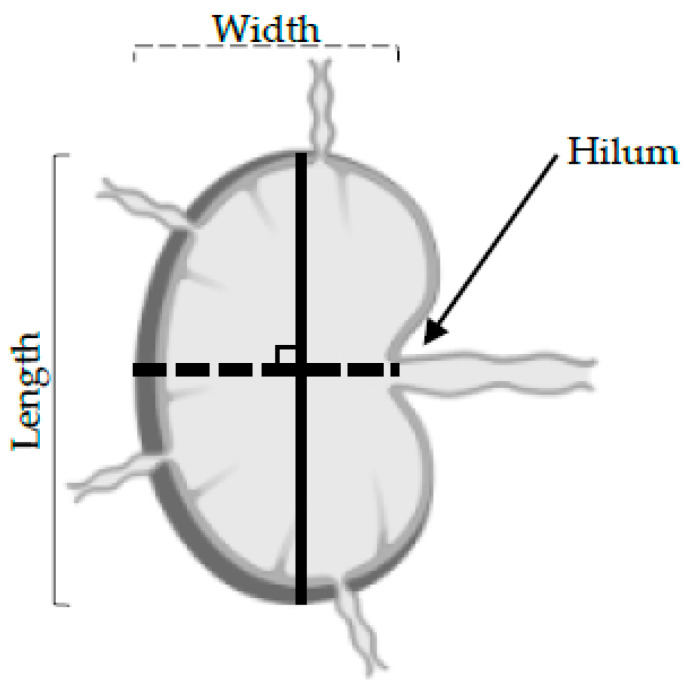
Schematic of a LN. Shape was determined by measuring the longest longitudinal diameter found on the 3D US data and 2D B-mode image acquired during the 3D reconstruction was inspected for the presence of a hilum.

**Figure 3 cancers-14-00561-f003:**
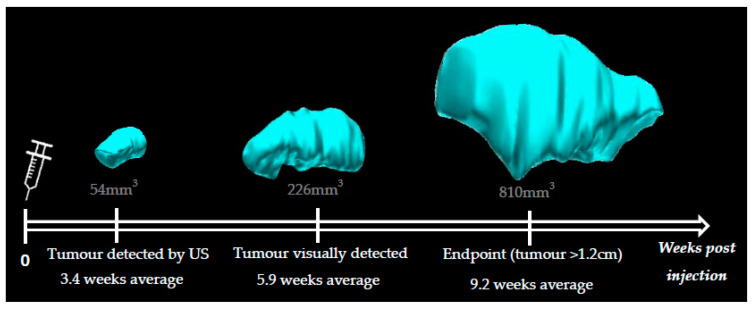
3D reconstitutions of the hind leg tumor of KPN-injected mouse 2 over time. The leg tumors from KPN-injected animals were detected by ultrasound from 2.5 weeks post organoids injection (3.4 weeks average) and visually detected from 4 weeks post injection (5.9 weeks average). Average endpoint was reached 9.2 weeks post organoid injection. Average size of the tumor at 3.4, 5.9, and 9.2 weeks was 54 mm^3^, 226 mm^3^, and 810 mm^3^, respectively.

**Figure 4 cancers-14-00561-f004:**
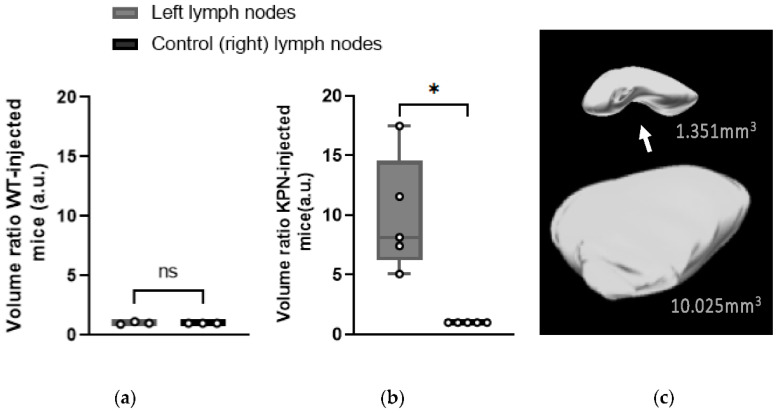
Volume ratio of the left inguinal LNs compared to the control (right) LNs at endpoint. (**a**) Mice injected with WT intestinal organoids (*n* = 3 mice); (**b**) mice injected with KPN intestinal tumor organoids (*n* = 5 mice). Paired *t*-test analysis. ns: *p* > 0.05; *: *p* < 0.05; (**c**) Representative 3D reconstitutions of right control (top image) and left tumor-draining (bottom image) inguinal LNs from KPN-injected animals at endpoint and their respective volumes. Arrow: hilum of the LN containing the efferent lymphatic vessels and blood vessels. No hilum was detected on the tumor-draining LN (bottom image).

**Figure 5 cancers-14-00561-f005:**
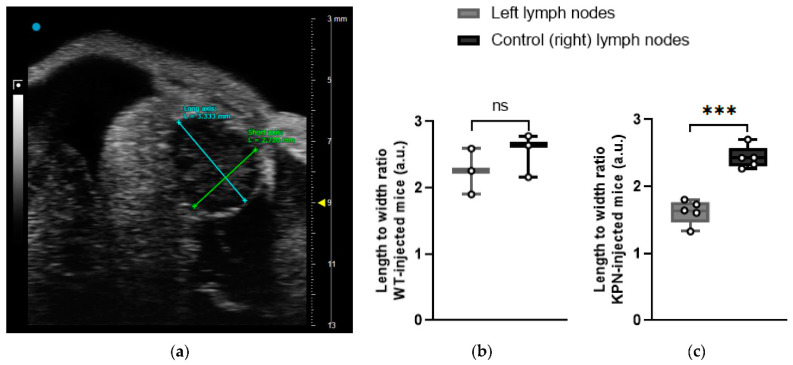
(**a**) 3D-US frame showing hypoechogenic tumor-draining inguinal LN. The long axis was defined as the longest diameter across all frames of the US data, and the short axis was determined as the longest perpendicular diameter from the same frame used for the long axis. (**b**) Length–width ratio of left and control (right) LNs of WT-injected animals is not significantly different (*n* = 3 mice); (**c**) length–width ratio of left and control (right) LNs of KPN-injected animals is significantly different (*n* = 5 mice). Paired *t*-test analysis. ns: *p* > 0.05; ***: *p* < 0.001.

**Figure 6 cancers-14-00561-f006:**
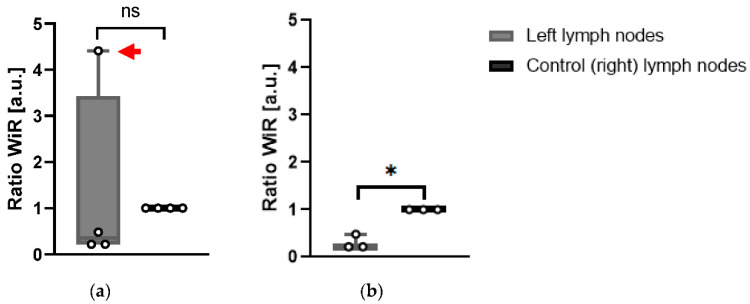
(**a**) Wash-in rate ratios of left LNs compared to control (right) inguinal LNs of KPN-injected mice. Red arrow: Mouse 4 tumor-draining LN WiR different from other left LNs (*n* = 4); (**b**) Wash-in rate ratios without Mouse 4 (*n* = 3) (extreme outlier-reason for exclusion indicated in Section 3, Section 4 and Section 5). Paired *t*-test analysis. ns: *p* > 0.05; * *p* < 0.05.

**Figure 7 cancers-14-00561-f007:**
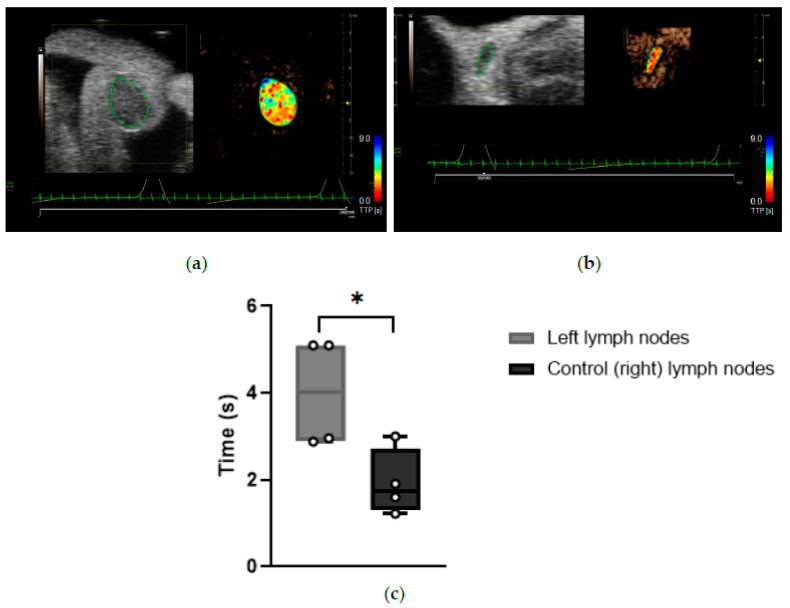
(**a**) Representative software analysis image showing time to peak of a tumor-draining inguinal LN. Heat maps represent the time the contrast agent took to reach peak enhancement. Blue: long TTP; red: short TTP. (**b**) Representative software analysis image showing time-to-peak of a control (right) inguinal LN. Heat maps represent the time the contrast agents took to reach peak enhancement. Blue: long TTP; red: short TTP. (**c**) Time-to-peak of tumor-draining and control (right) inguinal LNs (*n* = 4 mice). *: *p* < 0.05.

**Figure 8 cancers-14-00561-f008:**
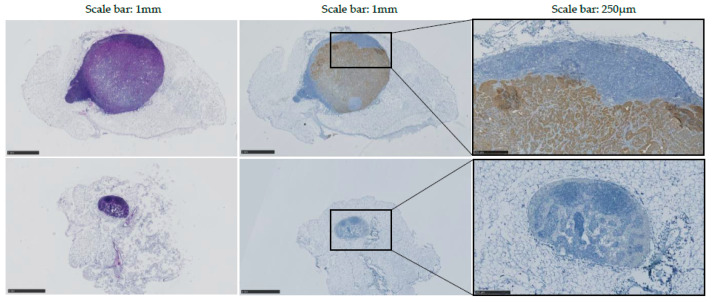
H&E and immunostaining of Mouse 1 KPN-injected LN; 80% (4/5) of KPN-injected mice presented detectable metastasis at endpoint. Top: left inguinal LN; bottom: control (right) inguinal LN. Left panels: H&E staining; middle and right panels: Cytokeratin 7 staining intestinal tumor in brown.

**Table 1 cancers-14-00561-t001:** Parameters used for the characterization of LNs and associated scoring system. For volume, wash-in rate, and time-to-peak, data from left LNs were compared to data from control right LNs.

Score	Volume Ratio	Shape	Hilum	Wash-in Rate	Time-to-Peak
Score: 1	Left LN at least 5× larger than control (right) LN	Length–width ratio < 2 (round)	Not detected	Significantly reduced in left LN compared to control (right) LN	Significantly increased in left LN compared to control (right) LN
Score: 0	Left LN smaller or equal to control (right) LN	Length–width ratio > 2 (non-round)	Detected	Lengthened in left LN or equal to control (right) LN	Reduced in left LN or equal to control (right) LN

**Table 2 cancers-14-00561-t002:** Scoring system for left inguinal LNs of KPN-injected mice (1, 2, 3, 4 and 5) and WT-injected mice (A, B and C) compared to their respective control (right) inguinal LNs. Scoring is indicated in Table 1. CEUS data of Mouse 5 could not be analyzed due to the software failure to perform satisfactory motion correction (N/A); consequently, the sum score for this mouse was done using only 3 parameters.

Parameter	Mice Injected with KPN Tumor Organoids					Mice Injected with WT Organoids		
	Mouse 1	Mouse 2	Mouse 3	Mouse 4	Mouse 5	Mouse A	Mouse B	Mouse C
Volume	1	1	1	1	1	0	0	0
Shape	1	1	1	1	1	1	0	0
Hilum	1	1	1	1	1	0	0	0
TTP	1	1	1	1	N/A	0	0	0
WiR	1	1	1	0	N/A	0	0	0
Sum score	5/5	5/5	5/5	4/5	3/3	1/5	0/5	0/5
Sum score (%)	100	100	100	80	100	20	0	0

**Table 3 cancers-14-00561-t003:** Specificity, positive predictive value (PPV), negative predictive value (NPV), and diagnostic accuracy for individual parameters or at different parameter sum scores (%). Sensitivity for every parameter and sum score was 100%; 95% confidence intervals (%) for those tests are in square brackets.

Variable	Specificity (%)	PPV (%)	NPV (%)	Diagnostic Accuracy
Individual parameters				
Volume	75 [19–99]	80 [28–99]	100 [29–100]	88 [47–100]
Shape	50 [7–93]	67 [22–96]	100 [16–100]	75 [35–97]
Hilum	75 [19–99]	80 [28–99]	100 [29–100]	88 [47–100]
TTP	75 [19–99]	75 [19–100]	100 [29–100]	86 [42–100]
WiR	100 [40–100]	100 [29–100]	100 [40–100]	100 [59–100]
Sum score				
0	50 [7–93]	67 [22–96]	100 [16–100]	75 [35–97]
20	50 [7–93]	67 [22–96]	100 [16–100]	75 [35–97]
40	75 [19–99]	80 [28–99]	100 [29–100]	88 [47–100]
60	75 [19–99]	80 [28–99]	100 [29–100]	88 [47–100]
80	75 [19–99]	80 [28–99]	100 [29–100]	88 [47–100]
100	100 [40–100]	100 [40–100]	100 [40–100]	100 [63–100]

## Data Availability

The data presented in this study are openly available at https://doi.org/10.15129/556fadaf-3068-484d-8971-836ccf6e0d63 (accessed on 2 January 2022).

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
