# Peer review of "Development of Preclinical Ultrasound Imaging Techniques to Identify and Image Sentinel Lymph Nodes in a Cancerous Animal Model"

_cancers, 2022, doi:10.3390/cancers14030561_

Round 1

Reviewer 1 Report

The authors have clarified the manuscript which is more understandable in its present form.

Reviewer 2 Report

The paper is resubmitted from cancers-1472341
Based on the previous reviewer comments, the authors updated the results, provided more details about individual parameters used in calculating the proposed score, enhanced the language to better describe the method and the results, fixed the figures, and enriched the discussion. 

This manuscript is a resubmission of an earlier submission. The following is a list of the peer review reports and author responses from that submission.

Round 1

Reviewer 1 Report

The manuscript submitted by Bacou et al., entitled "Development of preclinical ultrasound imaging techniques to identify and image sentinel lymph nodes in a cancerous animal model", presents an experimental study to predict the likelihood of lymph node drainage being involved in cancer, based on a scoring system created by US imaging.

In my opinion, this is a well-conceived article, based on a novel idea that, if demonstrated, could represent a turning point in the aggressive surgery of abdominal metastases. However, in its present form, the paper suffers from some shortcomings that preclude its publication.

Introduction

  • The introduction section is too brief. It lacks information on how research in the proposed line is currently being carried out.
  • In line 69 the authors write "we developed a metastatic mouse model", I am not sure if the model proposed is really newly created or not.

Animals & Cells

  • Line 78-79. Mice were housed in individually ventilated cages. The authors should be aware that individual housing of experimental animals induces varying degrees of stress to gregarious animals, such as mice, which may influence the experimental results. This should be taken into account in the discussion of the results.
  • Line 82 and 85. Number of animals. The authors assigned 5 animals to the KPN-injected group and 3 animals to the control group (WT-injected). The number of animals used is definitely insufficient, and even more so if, as the authors state in line 69, this is a work describing a new experimental model.
  • The organoids were induced by the authors through the administration of tamoxifen, so it is necessary to present a characterization of the cells present in these organoids, to ensure, without any doubt, that they contain intestinal tumor cells. The same applies to the conditioned in-house manufactured media; its composition must be provided since it is not a well-defined and characterized commercial medium.
  • Line 109. Why did the authors establish 1.2 cm diameter as a humane end-point criterion? Why that measurement and not a smaller or larger one?
  • Line 149. Figure 4 is cited in the text before figure 2, the figures should be named according to their order of appearance in the text of the manuscript.
  • Line 150 and others. There is an error in the citation of the tables, the reference source is not found.

  1. Results
  • The results presented are very promising and are very well described, however, and in general, they cannot be considered and the significance obtained cannot be accepted as the number of animals is very small.
  • Line 330 and 337. Figure 7 is duplicated in the manuscript.
  1. Discussion
  • The discussion is very well oriented, but, like the results section, in my opinion, it lacks validation due to the serious shortcoming related to the small number of animals used, and to other deficiencies in the methodology

Sorry for the inconvenience.

Yours faithfully,

Reviewer 2 Report

The authors proposed using ultrasound imaging to predict whether cancer has invaded the lymph nodes in the mouse models. The authors reported the results of studying 8 mouse models (5 infected with cancer and 3 normal controls). Multiple characteristics of the studied ultrasound images of lymph nodes including lymph node volume, shape, hilum, Wash-in rate and Time to peak have been reported. A score for predicting metastatic cancer in lymph nodes has been devised. The reviewer has the following comments regarding the work reported here,

- The authors claim that the combined score enhances the sensitivity of the cancer detection compared to individual features while the sensitivity of all individual features are 100%?

- The authors claim the the combined scores enhances accuracy while no accuracy metrics are provided for individual features or the combined score

- The combined score at 60% coexistence is equivalent to the volume feature (see table 3)

- It is not clear if any threshold on the volume difference between left and right nodes has been used. It may be useful to use the ratio or difference between the two nodes instead of having a binary feature

- The authors treated all the features equally in calculating their combined score. It may be useful to assess what feature is more important.

- It is not clear how the authors dealt with the missing values of WiR when calculating specificity, sensitivity, NPV and PPV

-Figure 4: use the same scale on y axis of a and b

-Figure 2 is repeated multiple times and it interferes with the text. The formating needs to be fixed

- Figure 1 can benefit from higher resolution or it can be embedded as vector image

- Error! Reference source not found.). In line 347

- Table 2. and Table 3 can be combined to provide better comparisons
